# Impact of Nutrient and Stoichiometry Gradients on Microbial Assemblages in Erhai Lake and Its Input Streams

**Yang Liu** [1,2,3] 🆔**, Xiaodong Qu** [1,3]**, James J. Elser** [4] 🆔**, Wenqi Peng** [1,3]**, Min Zhang** [1,3,*]**, Ze Ren** [4,*]**, Haiping Zhang** [1,3]**, Yuhang Zhang** [1] **and Hua Yang** [5]

1    State Key Laboratory of Simulation and Regulation of Water Cycle in River Basin, China Institute of Water Resources and Hydropower Research, Beijing 100038, China
2    College of Hydrology and Water Resources, Hohai University, Nanjing 210098, China
3    Department of Water Environment, China Institute of Water Resources and Hydropower Research, Beijing 100038, China
4    Flathead Lake Biological Station, University of Montana, Polson, MT 59860, USA
5    Yunnan Hydrology and Water Resources Bureau Dali Branch, Dali 671000, China
*    Correspondence: zhangmin@iwhr.com (M.Z.); renzedyk@gmail.com (Z.R.); Tel.: +86-010-6878-1946 (M.Z.)

**Abstract:** Networks of lakes and streams are linked by downslope flows of material and energy within catchments. Understanding how bacterial assemblages are associated with nutrients and stoichiometric gradients in lakes and streams is essential for understanding biogeochemical cycling in freshwater ecosystems. In this study, we conducted field sampling of bacterial communities from lake water and stream biofilms in Erhai Lake watershed. We determined bacterial communities using high-throughput 16S rRNA gene sequencing and explored the relationship between bacterial composition and environmental factors using networking analysis, canonical correspondence analysis (CCA), and variation partitioning analysis (VPA). Physicochemical parameters, nutrients, and nutrient ratios gradients between the lake and the streams were strongly associated with the differences in community composition and the dominant taxa. Cyanobacteria dominated in Erhai Lake, while Proteobacteria dominated in streams. The stream bacterial network was more stable with multiple stressors, including physicochemical-factors and nutrient-factors, while the lake bacterial network was more fragile and susceptible to human activities with dominant nutrients (phosphorus). Negative correlations between bacterial communities and soluble reactive phosphorus (SRP) as well as positive correlations between bacterial communities and dissolved organic carbon (DOC) in the network indicated these factors had strong effect on bacterial succession. Erhai Lake is in a eutrophic state, and high relative abundances of Synechococcus (40.62%) and Microcystis (16.2%) were noted during the course of our study. CCA indicated that nutrients (phosphorus) were key parameters driving Cyanobacteria-dominated community structure. By classifying the environmental factors into five categories, VPA analyses identified that P-factor (total phosphorus (TP) and SRP) as well as the synergistic effect of C-factor (DOC), N-factor ($NO_3^-$), and P-factor (TP and SRP) played a central role in structuring the bacterial communities in Erhai Lake. Heterogeneous physicochemical conditions explained the variations in bacterial assemblages in streams. This study provides a picture of stream–lake linkages from the perspective of bacterial community structure as well as key factors driving bacterial assemblages within lakes and streams at the whole watershed scale. We further argue that better management of phosphorus on the watershed scale is needed for ameliorating eutrophication of Erhai Lake.

**Keywords:** stream-lake linkage; taxonomic; bacterial community; environmental change

## 1. Introduction

Microbial assemblages are fundamental components of aquatic ecosystems and play a key role in biogeochemical cycles in both lotic and lentic ecosystems [1,2]. Their high diversity, small size, and rapid generation have caused microbes to become the most sensitive aquatic organisms to environmental perturbations, especially to nutrient alteration [3–5]. Understanding these responses is increasingly important given that biogeochemical cycles have been dramatically altered by human activities [6], with a two-fold increase in nitrogen (N) availability [7,8] and approximately a four-fold increase in phosphorus (P) mobilization [9,10] relative to preindustrial times.

Freshwater ecosystems include networks of lakes and streams linked within catchments by downslope water flow and nutrient flux [11]. The spatial distribution of lakes and streams can influence various chemical and biological characteristics, while the interactions among aquatic habitats control, at least in part, their individual functioning [11,12]. Ecosystem metabolism is directly influenced by the relative balance of external loading of nutrients and dissolved organic carbon in lakes [13]. Stream ecosystems are primary receivers of nutrients and organic matter that export from terrestrial ecosystems through transport and storage of water, nutrients, and energy [14–16], and these nutrients and organic matter are transported by streams into lakes and marine environments [13]. In stream ecosystems, microbial assemblages primarily occur in the form of benthic biofilms [17], which play an important role in biogeochemical cycling, being responsible for organic matter mineralization, nutrient uptake, and transformation of contaminants [18]. Likewise, bacterial communities in lake ecosystems often have a close relationship with nutrient alteration. Data currently available suggest that bacterial community composition in freshwater ecosystems is controlled both by biogeography and environmental nutrient supplies, and that the relative importance of these controls shifts spatially from local to regional areas [11,19,20]. However, most studies of bacterial communities have focused only on single environment types, such as lakes or streams. Comparative information on the comparison of bacterial assemblages and their driving forces in lakes and input streams is still limited [16].

Erhai Lake is the second largest plateau lake and the seventh largest freshwater lake in China. This lake has valuable functions in supporting human life and regional development, including drinkable water supply, agricultural irrigation, climate regulation, tourism, and hydroelectric power generation [21–23]. Anthropogenic activities in the basin generate a large amount of nutrients and waste, which directly perturb biogeochemical cycles and accelerate whole-basin ecosystem deterioration [24]. Previous research on the bacterial communities in Erhai Lake basin focused only on streams [25] or lakes [26–29]. At the watershed scale, the ecosystems of Erhai Lake and its tributaries are highly connected by material and energy fluxes. To gain a better understanding of how nutrients are associated with variations in bacterial communities between the two kinds of ecosystems, we conducted an extensive bacterial survey using high-throughput ILLUMINA sequencing of 16S rRNA in Erhai Lake and its surrounding major tributaries. This study aimed to address four major questions regarding Erhai Lake and streams: (1) How do they differ in bacterial community structure? (2) How are co-occurrence patterns of bacterial taxa structured? (3) What are the major environmental factors associated with bacterial assemblages at the watershed scale? (4) How do these associations differ for the lake and the inflow streams? By understanding bacterial communities and its associations with human activities in the lake and its input streams, this study seeks to provide insight into the management of Erhai Lake watershed.

## 2. Materials and Methods

### 2.1. Study Area

Erhai Lake (25°25′–26°16′ N, 99°32′–100°27′ E) is the second largest freshwater lake in China. Erhai Lake watershed experiences a typical subtropical plateau monsoon climate with 85% of precipitation occurring from May to October and long-term average annual precipitation of 909 mm. The annual average air temperature is 15.7 °C (22 °C on average in summer). Erhai Lake watershed is located

in the upper Mekong River tributaries with a total of 117 tributaries. Miju stream is the largest tributary of Erhai Lake originating in the northern mountains of Erhai Lake watershed, and there are also many small tributaries located in the western part of Erhai Lake. All of these tributaries pass through urbanized areas around the lake, which dramatically altered physical, chemical, and biological conditions of stream ecosystems. Moreover, the rapid urbanization around the lake has caused severe lake eutrophication and water contamination [27].

### 2.2. Sampling and Physicochemical Analyses

During August 2016, a total of nineteen samples spread evenly across Erhai Lake were collected, and seventeen samples were collected from six tributaries [Miju stream (MJ), Wanhuaxi stream (WH), Yangxi stream (YX), Jinxi stream (JX), Baishi stream (BS), and Taoxi (TX) stream]. According to the different level of human activities between upstream and downstream, two sampling sites were chosen in each stream.. Surface water samples for bacterial analyses were collected at a depth of 0.5 m in Erhai Lake. A total of 600 mL water was filtered with Whatman Nylon membrane filters (pore size: 0.2 μm), and these filters were immediately stored and transported in liquid $N_2$. Benthic biofilms were removed by vigorously scrubbing (with a sterilized nylon brush) a 4.5 cm-diameter area from the surface of a stone substrate with five replicates at each sampling site. After the slurry was rinsed with sterile water, 10 mL of the mixed slurry was filtered with 0.2 μm membrane filters, and these filters were immediately frozen in liquid $N_2$ in the field. All bacterial samples were stored at −80 °C until DNA extraction and subsequent analyses.

At each sampling site, water temperature (Temp), dissolved oxygen (DO), pH values, and conductivity (Cond) were measured in situ using a YSI Model 80 m (Yellow Springs Instruments, Yellow Springs, OH, USA). Altitude was measured using a Global Positioning System (GPS) unit (Triton 500, Magellan, Santa Clara, CA, USA). Water samples were acid-fixed and transported to the laboratory at 4 °C for chemical analyses. Water was filtered through Whatman Glass Fibre Filters to analyze dissolved organic carbon (DOC), soluble reactive phosphorus (SRP), nitrate ($NO_3^-$), and ammonium ($NH_4^+$). Total nitrogen (TN) was analyzed using ion chromatography after persulfate oxidation. $NO_3^-$ was determined using ion chromatography, and $NH_4^+$ was analyzed using the indophenol colorimetric method. Total phosphorus (TP) and SRP were quantified using the ammonium molybdate method. DOC was analyzed using a Shimadzu TOC Analyzer (TOC-VCPH, Shimadzu Scientific Instruments, Columbia, MD, USA).

### 2.3. DNA Extraction, PCR, and Sequencing

The bacterial community was characterized by 16S rRNA gene sequencing. Bacterial genomic DNA was extracted using a PowerSoil DNA Isolation Kit (MoBio, Carlsbad, CA, USA) following the manufacturer's protocols. The 16S rRNA genes covering the V3 to the V4 regions were amplified using primers 806R-GGACTACHVGGGTWTCTAAT and 338F-ACTCCTACGGGAGGCAGCA (Invitrogen, Vienna, Austria) [25]. The PCR was performed according to the standard procedures of Applied Biosystems 2720 Thermal Cycler (ABI, Foster City, CA, USA). By using 1× Tris-Acetate-Edta buffer, amplified DNA samples were verified by 1.0% agarose gel electrophoresis and purified using the Gel Extraction Kit (Qiagen, Hilden, Germany). The final sequencing process was based on a MiSeq sequencing platform (Illumina, San Diego, CA, USA).

Raw sequence data (available from the National Center for Biotechnology Information, SRP167734) were processed using QIIME software [30]. The range of sequence reads per sample was 12,009–30,315. Quality filtering on merged sequences was performed. Sequences were discarded when these sequences did not meet the following criteria: sequence length < 200 bp, no ambiguous bases, and mean quality score > 20. Then, these sequences were compared with reference database (RDP Gold database) using the UCHIME algorithm to detect chimeric sequences [31], and the chimeric sequences were removed. After quality filtering, these sequences were grouped into operational taxonomic units (OTUs) using the clustering program VSERACH 1.9.6 [32] against the Silva 123 database based on the complete

linkage algorithm at an identity level of 97% sequence similarity. The Ribosomal Database Program (RDP) classifier was used to assign a taxonomic category to all OTUs at a confidence threshold of 0.8. The RDP classifier uses the Silva 123 database, which has taxonomic categories predicted to the species level.

*2.4. Data Analysis*

We compared taxonomic profiles of the bacterial communities to assess the differences and the linkages between bacterial communities in lake and streams. To determine whether physicochemical factors, nutrients, organic carbon, and stoichiometric factors were significantly different between lake and streams, we conducted bootstrap *t*-tests using SPSS software (Version 12.0) (IBM, Armonk, NY, USA). The $DOC:NO_3^-:TP$ (C:N:P) ratios were calculated based on stoichiometric homeostasis [33] with the following equation:

$$(x:y)_{\text{molar}} = \frac{x}{y} \frac{y_n}{x_n}$$

where $x$ and $y$ are nutrient concentrations, and $x_n$ and $y_n$ are the molar mass.

The spatial variability of nutrient concentrations, nutrient ratios, and bacterial community composition was visualized in ArcGIS 10.2 (ESRI, Redlands, CA, USA). The Multiple Response Permutation Procedure (MRPP) is a nonparametric procedure of testing the null hypothesis between two or more groups of entities in the software R (version 3.3.2) (Lucent Technologies, Murray Hill, NJ, USA). Heatmaps were applied to reveal the differences of OTUs (relative abundance higher than 0.1%, Heatplus and Gplots packages in R version 3.3.2). All pairwise Spearman's rank correlations between those OTUs and environmental variables were calculated in R (Hmisc package, version 4.0-1). Only robust ($r > 0.6$ or $r < -0.6$) and statistically significant (*p*-values $< 0.01$, *p*-values were adjusted using Benjaminin–Hochber method) were considered. Cytoscape (version 3.5.1) was used for network visualization. Nodes represented the OTUs, and edges represented Spearman's correlation relationship. All of the nodes were classified at the phylum level, and an edge-weighted spring-embedded network was applied to display the arrangement of the variables. Topological and node/edge metrics, including centralization, heterogeneity, characteristic path length, and clustering coefficient, were calculated. Modular structure analyses were conducted using the ClusterMaker app in Cytoscape. A total of 30 random networks with the same size of a real network were generated using Network Randomizer app (version 1.1.3) and topological parameters were calculated individually. The *t*-tests were employed to calculate the differences between topological parameters of the real network and the random networks using the standard deviations derived from the corresponding random network. Canonical correspondence analysis (CCA) was applied to analyze the bacterial communities with respect to various environmental factors using Vegan package 2.4 in R (version 3.3.2). Monte Carlo permutations and Mantel tests ($p < 0.05$) were used to select a set of environmental variables that had significant and independent effects on bacterial communities. Environmental factors with high partial correlation coefficients ($p < 0.05$, $r > 0.5$) and variance inflation factors $> 20$ were eliminated from the final CCA. To identify effects of different categories of environmental variables on bacterial communities, environmental factors were classified into five categories (Table 1). Variation partitioning analysis (VPA) was performed using the Vegan package (version 2.5-3) in R to determine the relative contributions of physicochemical-factor, N-factor ($NO_3^-$), P-factor (TP and SRP), C-factor (DOC), and the interactions of various factors.

**Table 1.** Environmental variables in Erhai Lake and streams.

| Categories of Environmental Variables | Parameters | Lake | | | Stream | | | $p$ (*t*-tests) |
|---|---|---|---|---|---|---|---|---|
| | | Average | SD | CV | Average | SD | CV | |
| Physicochemical-factor | Altitude (m) | 1964.00 | 0.00 | 0 | 2200.28 | 265.28 | 0.071 | 0.001 ** |
| | Temp (°C) | 22.93 | 0.90 | 0.039 | 16.68 | 1.99 | 0.121 | 0.000 ** |
| | DO (mg/L) | 5.50 | 0.44 | 0.079 | 5.10 | 0.59 | 0.120 | 0.077 |
| | Cond (μs/cm) | 287.28 | 6.13 | 0.021 | 115.29 | 52.46 | 0.738 | 0.000 ** |
| | pH | 9.53 | 0.07 | 0.007 | 8.36 | 0.82 | 1.537 | 0.000 ** |
| P-factor | TP (mg/L) | 0.04 | 0.01 | 0.146 | 0.06 | 0.02 | 0.327 | 0.011 * |
| | SRP(mg/L) | 0.01 | 0.002 | 0.267 | 0.028 | 0.008 | 0.431 | 0.000 ** |
| N-factor | TN (mg/L) | 0.54 | 0.08 | 0.138 | 0.53 | 0.32 | 0.529 | 0.863 |
| | $NO_3^-$ (mg/L) | 0.50 | 0.06 | 0.125 | 0.47 | 0.29 | 0.565 | 0.690 |
| | $NH_4^+$ (mg/L) | 0.017 | 0.004 | 0.259 | 0.016 | 0.003 | 0.471 | 0.621 |
| C-factor | DOC (mg/L) | 4.95 | 0.44 | 0.088 | 1.16 | 0.25 | 0.437 | 0.000 ** |
| Nutrient ratios | $DOC:NO_3^-$ (C:N) | 11.626 | 1.282 | 0.110 | 4.366 | 2.597 | 0.595 | 0.001 ** |
| | DOC:TP (C:P) | 293.011 | 46.494 | 0.158 | 79.068 | 26.261 | 0.332 | 0.000 ** |
| | $NO_3^-$:TP (N:P) | 25.267 | 3.400 | 0.135 | 24.119 | 14.978 | 0.621 | 0.761 |

** indicates $p < 0.01$ and * indicates $p < 0.05$. SD: standard division, CV: coefficient of variation, DO: dissolved oxygen, Cond: conductivity, TP: total phosphorus, SRP: soluble reactive phosphorus, TN: total nitrogen, DOC: dissolved organic carbon.

## 3. Results

### 3.1. Environmental Conditions and Nutrients

Significant differences in various environmental variables and nutrient concentrations were observed. Environmental variables were divided into five categories, including physicochemical-factor, P-factor, C-factor, N-factor, and nutrient ratios (Table 1). Temp, Cond, pH, and DOC were significantly higher in Erhai Lake than in streams (Table 1, *t*-tests, $p < 0.01$). TP and SRP concentration were higher in streams than in the lake ($p < 0.01$). Streams were the major sources of nitrogen and phosphorus in Erhai Lake (Figure 1). The concentration of TP in MJ was higher than that in other tributaries (*t*-tests, $p < 0.01$), which caused P concentrations to be higher in northern areas of the lake than in central and southern areas. Although SRP was the major component of TP in streams (29.17~74.44%), it only was a small component in the lake (15.38~30.95%). Moreover, $NO_3^-$ was the major component of TN in streams (57.09~93.78%) and the lake (86.18%~96.54%). In order to ensure the validity of data, we chose DOC, $NO_3^-$, and TP to calculate $DOC:NO_3^-$:TP (C:N:P). Both C:N and C:P ratios were higher in the lake than in streams (Table 1).

### 3.2. Bacterial Assemblages

After quality filtering, a total of 12,566 OTUs were identified. The relative abundance of different phyla dramatically differed in the lake relative to the streams. The proportion of shared OTUs was 27.9% between lake and streams, while the proportions of detected OTUs unique to Erhai Lake and streams were 31.1% and 41.0%, respectively. Except for the Bacteroidetes, all the other dominant phyla were significantly different between Erhai Lake and the streams. In the lake, the dominant phylum was Cyanobacteria (relative abundance of 43.8%), followed by Proteobacteria (15.1%) and Actinobacteria (14.8%) (Figure 2). The dominant phyla in streams were Proteobacteria (52.4%) and Cyanobacteria (30.9%) (Figure 2). Relative abundances of Proteobacteria and Thermi in streams were much higher than those in the lake (Figure 2, $p < 0.05$). Relative abundances of Cyanobacteria, Actinobacteria, Bacteroidetes, Verrucomicrobia, Chlorobi, and Planctomycetes were much higher in the lake than those in the streams (Figure 2, $p < 0.05$).

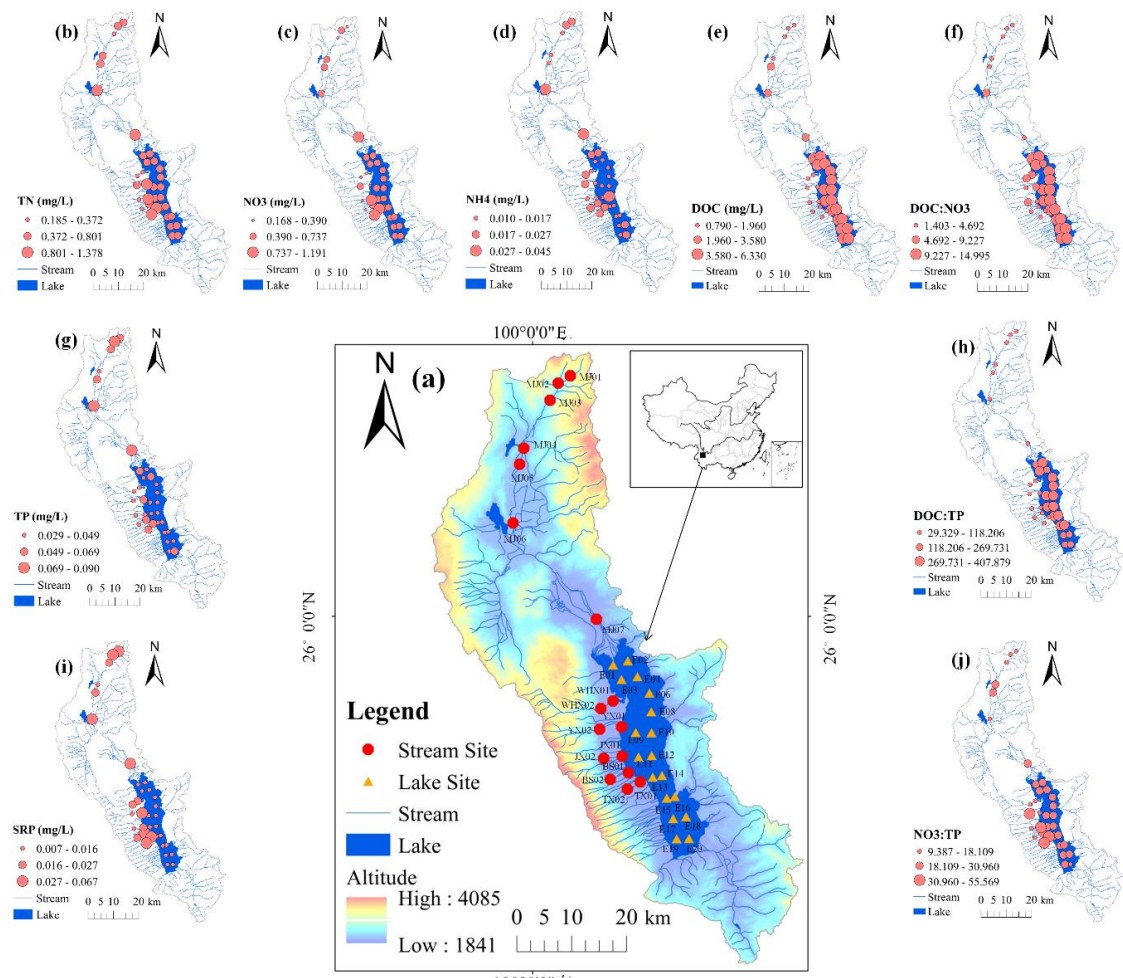

**Figure 1.** Study area and nutrient variables. (**a**) The distribution of sampling sites and (**b–j**) spatial patterns of nutrient variables in Erhai Lake and its tributaries. The map was created in ArcGIS 10.2 (http://desktop.arcgis.com/en/arcmap/) using ASTER GDEM data downloaded from the United States Geological Survey [ASTER GDEM is a product of the Ministry of Economy, Trade and Industry (METI) and the National Aeronautics and Space Administration (NASA)].

Unweighted Pair Group Method with Arithmatic Mean (UPGMA) clustering analyses illustrated the remarkable variations in bacterial community composition among the sampling sites (Figure 3a). In the lake, bacterial assemblages were clearly separated into two subgroups, and MRPP analyses demonstrated the significant difference between these two subgroups ($p < 0.05$). Subgroup 1 (EH09– EH11, EH14–EH18) had no dominant bacterial community under low nutrient concentrations (Figure 3a,b). In subgroup 2 (EH01–EH08, EH12, EH13, EH19, and EH20), Cyanobacteria (47.99–67.10%) was the dominant phyla under high phosphorus concentrations (Figure 3a,b). Bacterial community structure varied with spatial heterogeneity of environmental variables in streams. Heatmap clustering analyses also illustrated the variability of OTUs among sampling sites (Figure 4). Synechococcus (40.62%) and Microcystis (12.06%) made up the Cyanobacteria in Erhai Lake. Synechococcus (26.3%) dominated Cyanobacteria in streams, while Microcystis (0.23%) only was a small component of Cyanobacteria in streams.

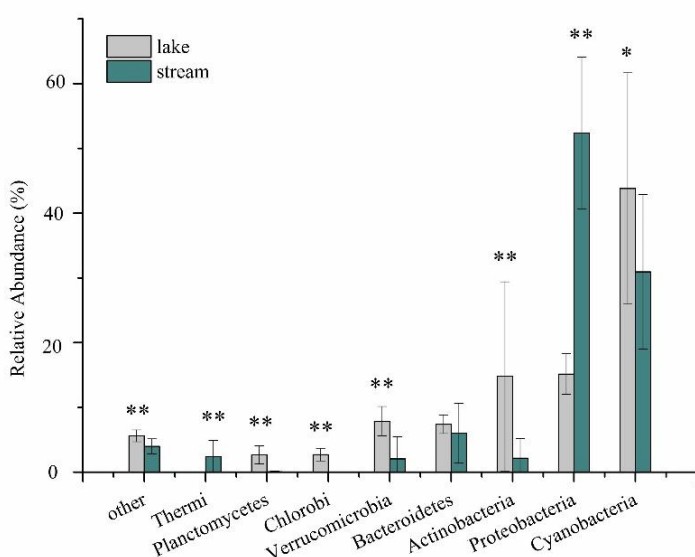

**Figure 2.** Phyla-level bacterial relative abundances in Erhai Lake and streams. Only phyla with a relative abundance > 1% in either streams or the lake are shown; "other" represents the unsigned operational taxonomic units (OTUs) and the phyla with a relative abundance < 1%. The comparison of lake and streams bacterial abundances was assessed by *t*-tests. ** indicates statistical significance at *p* < 0.01 and * indicates *p* < 0.05.

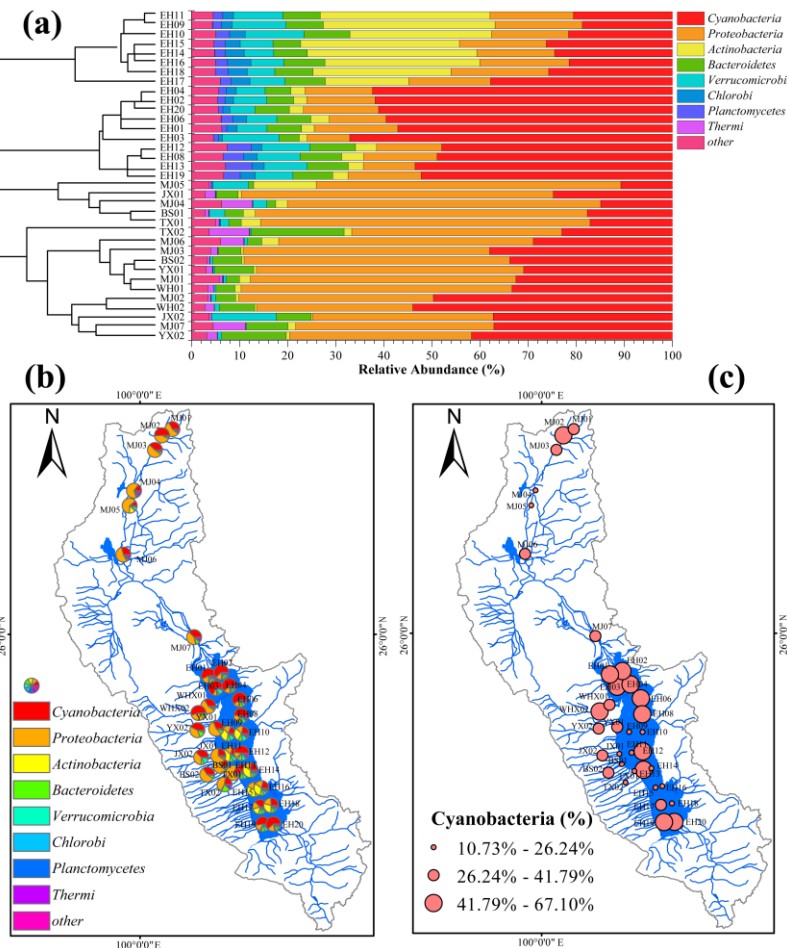

**Figure 3.** Composition of bacterial communities in Erhai Lake and streams (**a**) as well as the spatial variations in bacterial communities (**b**) and Cyanobacteria (**c**).

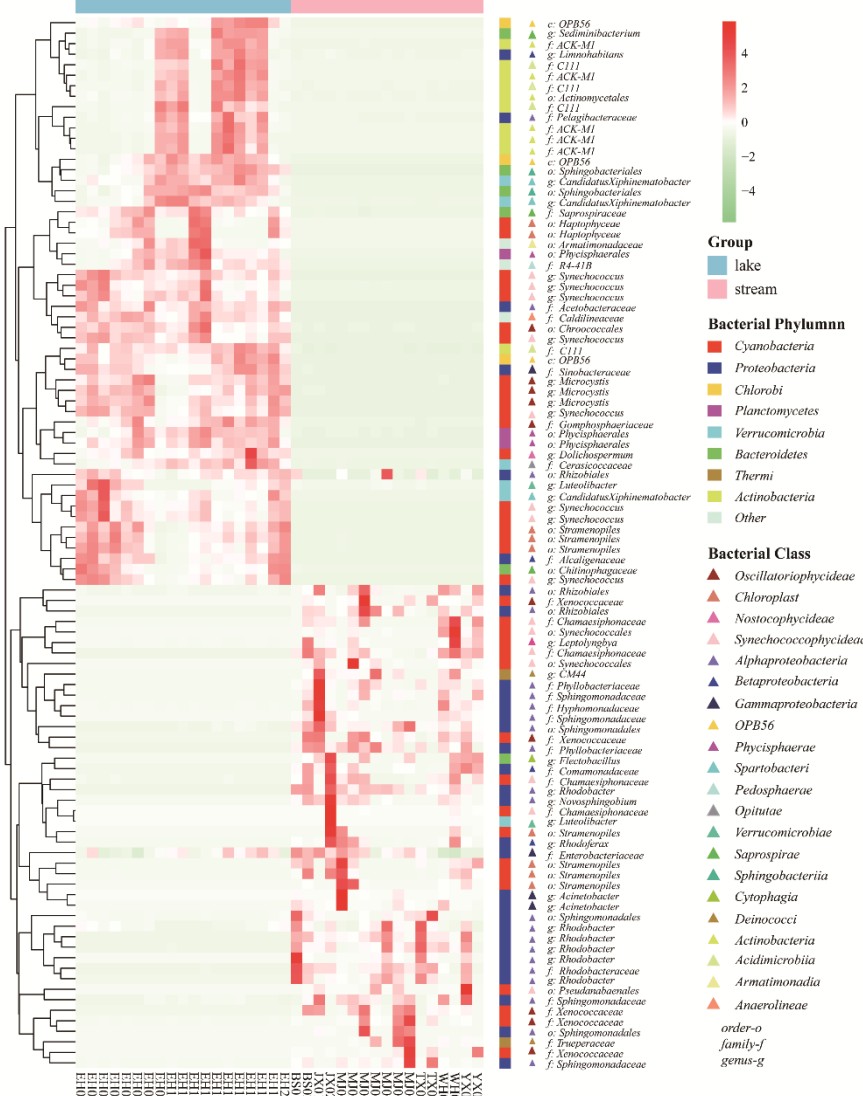

**Figure 4.** Heatmap showing the taxonomic differences of OTUs between Erhai Lake and streams based on Bray–Curtis distance. Bray–Curtis distances were calculated using relative abundances of OTUs (relative abundance > 0.1%).

### 3.3. Bacteria Co-Occurrence

OTUs (relative abundances > 0.01%) were used to build co-occurrence networks (Figure 5, $r > 0.6$ or $r < -0.6$, $p < 0.01$). The integrated bacterial network contained 511 nodes (i.e., OTUs) and 12 environmental parameters with 17,129 edges (significant interactions) (Figure 5a). The lake bacterial network contained 416 nodes (i.e., OTUs) and 5898 edges (Figure 5b), and the stream bacterial network contained 435 nodes with 2808 edges (Figure 5c). In total, 97.78% inter-genus correlations were positive within the integrated bacterial network. Significant correlations were mostly concentrated within Cyanobacteria, Proteobacteria, Verrucomicrobia, and Actinobacteria phyla. The positive relationship of OTUs in the lake bacterial network was 69.2% (Figure 5b), and in the streams network, it was 100% (Figure 5c). These results suggested that lake microorganisms have more competitive relationships than stream microorganisms due to the homogeneous habitat and the limited resources in the lake. In the integrated bacterial network, Proteobacteria had the highest accumulated betweenness centrality score of 0.67. The betweenness centrality is a measure of centrality of nodes in a network based on shortest paths. The higher the betweenness centrality scores of nodes are, the more associations over the network there are. C:P had the highest accumulated betweenness centrality score (0.065), followed

by SRP (0.035), Temp (0.033), and DOC (0.032). Moreover, both C:P (31.76%) and DOC (46.58%) were positively associated with nodes belonging to Cyanobacteria, while SRP (36.62%) was negatively associated with nodes belonging to Cyanobacteria.

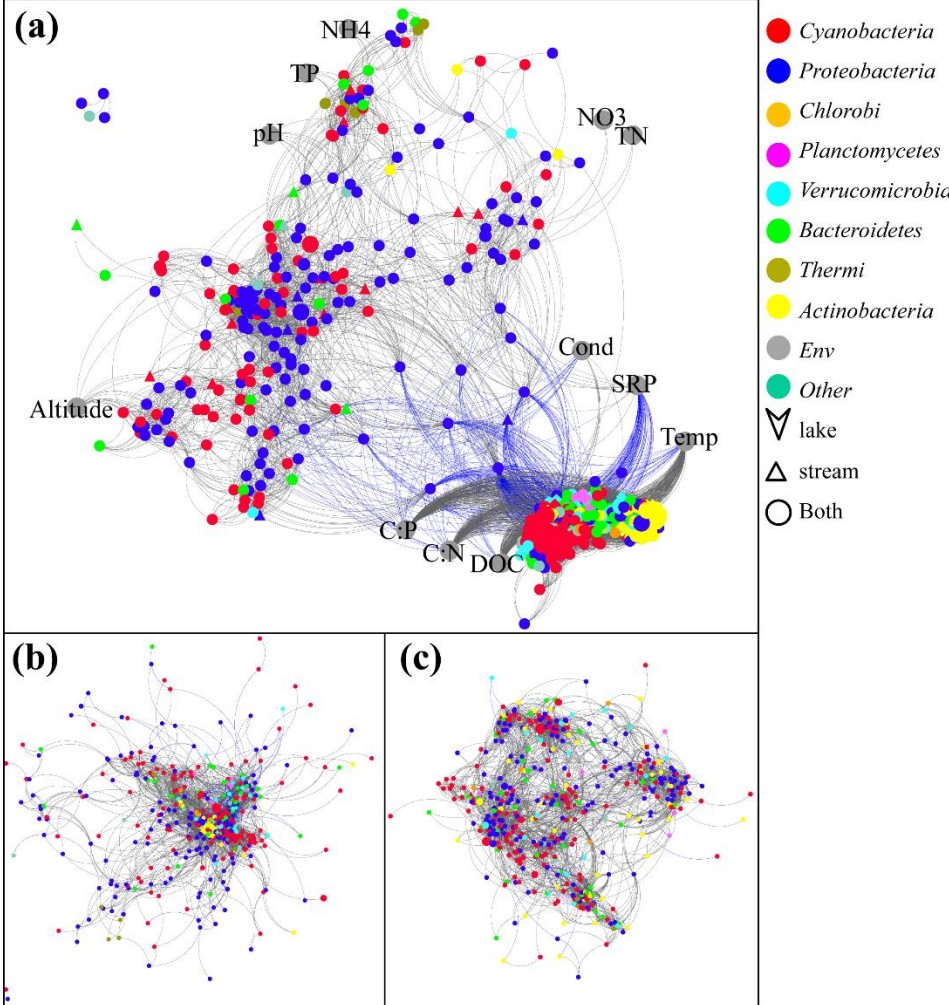

**Figure 5.** Co-occurrence networks of bacterial communities in integrated lake/stream (**a**), Erhai Lake (**b**), and streams (**c**). Nodes represent the OTUs. Edges represent Spearman's correlation relationships, and only strong and significant correlations (Spearman's $r > 0.6$ or $r < -0.6$, $p < 0.01$) are shown. The grey and the blue lines indicate positive and negative correlations, respectively.

Topological parameters were calculated to describe the interactions, the compactness, and the modularity among OTUs in the networks (Table 2). Among the three networks, the integrated lake/stream network showed the highest clustering coefficient and network centralization, while characteristic path length and modularity values were highest in the streams sub-network (Table 2). Bacterial communities in the three bacterial networks exhibited modular structures with modularity values ranging from 0.572 to 0.784 (Table 2). These results confirmed that all of these bacterial networks had "small world" properties and significant modular structures, because modularity, network centralization, clustering coefficient, and average path length were higher in real networks than in random networks (*t*-tests, $p < 0.05$).

**Table 2.** Topological parameters of real co-occurrence networks and corresponding random networks (some parameters of the random network are shown in mean ± SD. SD represent standard deviation). *t*-tests were used to compare the topological parameters between real and random networks.

| Topological Parameter | Lake and Streams | | Lake | | Stream | |
|---|---|---|---|---|---|---|
| | Random | Real | Random | Real | Random | Real |
| Number of nodes | 511 | 511 | 416 | 416 | 435 | 435 |
| Average number of neighbors | 65.840 | 67.037 | 28.351 | 28.351 | 12.906 | 12.906 |
| Network Centralization | 0.045 ± 0.003 * | 0.232 | 0.038 ± 0.004 * | 0.193 | 0.028 ± 0.008 * | 0.044 |
| Network Heterogeneity | 0.116 ± 0.004 * | 0.745 | 0.182 ± 0.006 * | 1.128 | 0.272 ± 0.001 * | 0.700 |
| Characteristic Path Length | 1.880 ± 0.001 * | 2.654 | 2.060 ± 0.001 * | 3.799 | 2.651 ± 0.002 * | 5.134 |
| Clustering Coefficient | 0.125 ± 0.006 * | 0.705 | 0.069 ± 0.001 * | 0.538 | 0.029 ± 0.002 * | 0.596 |
| Modularity | 0.439 ± 0.021 * | 0.639 | 0.438 ± 0.011 * | 0.572 | 0.446 ± 0.018 * | 0.784 |

* indicates *p* < 0.05.

### 3.4. Environmental Factors Associated with the Bacterial Community

Spearman's rank correlation, CCA, and VPA were used to assess the potential effect of environmental variables on bacterial communities (Figures 6 and 7). C:P was significantly associated with DOC and TP, N:P was significantly associated with $NO_3^-$ and TP, and C:N was only significantly associated with $NO_3^-$ (Figure 6). We noted that both C:P and N:P showed the co-limitation of the numerator and the denominator, while C:N was limited by the denominator in Erhai Lake. In streams, both C:P and N:P were limited by the numerator, and C:N was limited by the denominator, because DOC significantly associated with C:P, and $NO_3^-$ significantly associated with N:P and C:N (Figure 6). Moreover, TP, SRP, and C:P ratios were the predominant environmental factors closely associated with Cyanobacteria in the lake, and C:P ratios were also positively correlated with Verrucomicrobia (*p* < 0.05, Figure 6a). In streams, TN and $NO_3^-$ were positively associated with Cyanobacteria, and N:P ratios were negatively associated with Cyanobacteria (*p* < 0.05, Figure 6b). Moreover, Actinobacteria positively associated with TN, $NO_3^-$, DOC, and N:P (*p* < 0.05).

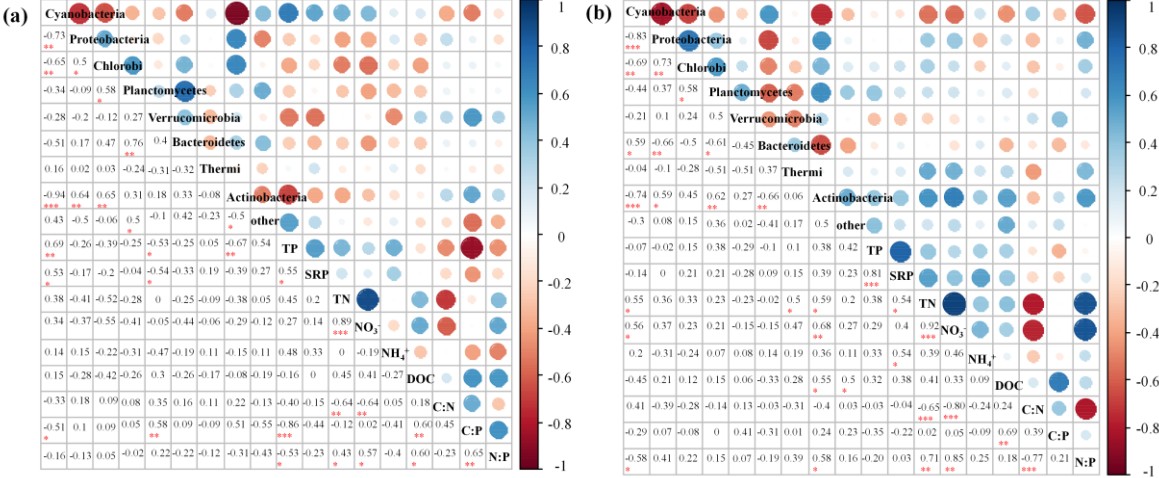

**Figure 6.** Spearman's rank correlation between nutrients and microbes at the phyla-level in (**a**) Erhai Lake and (**b**) streams. Blue nodes indicate positive associations and red nodes negative associations. Nodes are sized and present the degree of correlations. *** indicates statistical significance at *p* < 0.001, ** indicates *p* < 0.01 and * indicates *p* < 0.05.

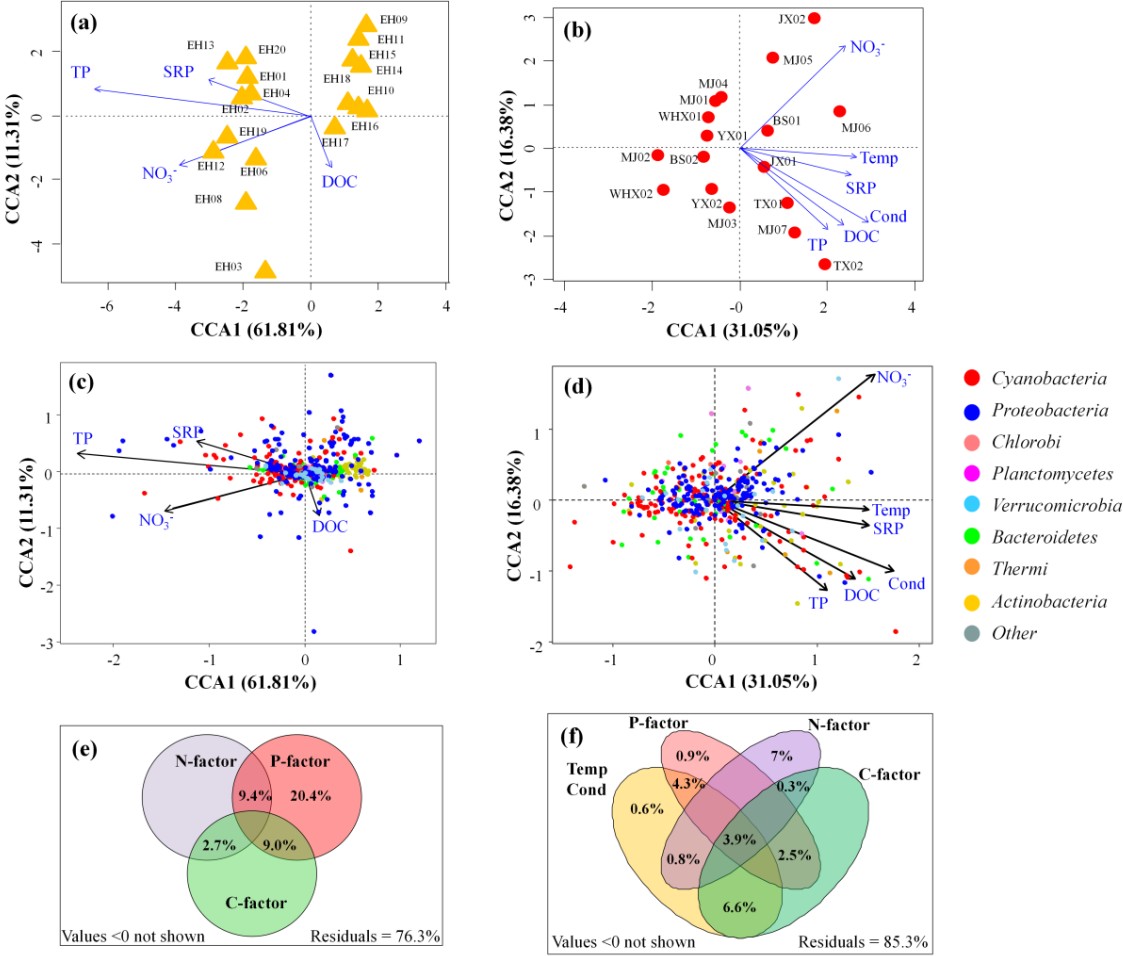

**Figure 7.** Canonical correspondence analysis (CCA) of bacterial communities and nutrient variables in Erhai Lake (**a, c, and e**) and streams (**b, d, and f**). All environmental variables had goodness of fit at the significant level $p < 0.05$ by envfit function. Variance partition analysis (VPA) determined the relative contributions of physicochemical factor, C-factors (DOC), N-factor ($NO_3^-$), P-factor (TP and SRP), and the interactions of these factors in Erhai Lake (**e**) and streams (**f**).

CCA and VPA were used to evaluate the relationship of environmental variables and bacterial communities (Figure 7). Some collinear variables were removed from CCA and VPA. $NO_3^-$ was the predominant N-factor used in CCA and VPA, because $NO_3^-$ was the major component of TN in Erhai watershed, and it positively associated with TN ($p < 0.01$, Figure 6). SRP did not dominate P-factor because SRP was a small component of TP in Erhai Lake; thus, TP and SRP constituted P-factor.

According to the Monte Carlo test and the Mantel test, TP, SRP, $NO_3^-$, and DOC were significantly associated with bacterial communities in Erhai Lake (Figure 7, two axes accounted for 73.12%, $p < 0.05$). The first axis was mainly defined by TP, SRP, and $NO_3^-$ in the negative direction, and the second axis was mainly defined by DOC in the negative direction. Following an increase of dominant nutrient parameters, OTUs gradually changed from Actinobacteria-dominated to Cyanobacteria-dominated (Figure 7c). The sampling sites were divided into two subgroups at the first axis. The subgroup 1 (EH09–EH11 and EH14–EH18), located at the right of the diagram, primarily involved samples from middle to southern parts of the lake, while the sites of subgroup 2 (EH01–EH08, EH12, EH13, EH19, and EH20), located at the left of the diagram, included samples from the northern areas along with two sites near the lake mouth to the down streams (Figure 7a). Variance partitioning analyses (VPA) revealed that nutrient factors explained 23.7% of bacterial communities in Erhai Lake. Moreover, P-factor and the two interactions of C, N, and P (C × P and N × P) had high contributions to the overall

explanation (Figure 7e). In addition, nutrients ratios (C:N, C:P, and N:P) explained 12.3% of bacterial communities in the lake (Figure S1).

Temp, Cond, $NO_3^-$, TP, SRP, and DOC were significantly associated with bacterial communities in streams (Figure 7b, two axes accounted for 47.43%, $p < 0.05$), and these variables defined the first axis. Bacterial communities were more evenly distributed, and no obvious pattern was identified (Figure 7d). Samples were divided into two subgroups based on axis 1 (Figure 7b): subgroup 1 (JX01, JX02, TX01, TX02, BS01, and MJ05–MJ07) and subgroup 2 (BS02, YX01, YX02, TX01, WHX01, and MJ01–MJ04). The samples of subgroup 1 (located at the right of the biplot) mainly represented sites experiencing higher levels of human disturbance or reference conditions in the streams, while the left direction represented the low levels of human activity (Figure 7b). N-factor, the interaction of physiochemical factor, C-factor, N-factor, and P-factor, had high contributions to the overall explanation of bacterial communities in the streams (Figure 7f). Nutrient ratios only explained 7.4% of bacterial communities in streams (Figure S1).

## 4. Discussion

For lacustrine bacterial assemblages, cells inflowing from tributaries can influence community structure [11,34]. These inputs partly explain the high level of overlap in identified OTUs between Erhai Lake and its input streams. However, according to similarity analyses of the bacterial communities, there was obvious spatial heterogeneity in the bacterial community structure between the lake and the streams, reflecting significant changes in relative abundances of various taxa. These results are consistent with previous studies of stream bacterial communities in connected lake and stream ecosystems [16,35]. The dramatic changes in bacterial community composition between the lake and the streams indicates significant variations in ecological function and nutrient metabolism [36]. In contrast to a previous study [26], our results showed extremely high relative abundances of Cyanobacteria in Erhai Lake. These high relative abundances of Cyanobacteria differed across sampling seasons, as Cyanobacteria always dominated Erhai Lake during summer [37]. We also found high abundances of Cyanobacteria in streams, while the dominant species of Cyanobacteria in streams were different from that in the lake. Because most of the Cyanobacteria are photoautotrophic, the increase in Cyanobacteria would likely correlate with changes in bacterial metabolism and biogeochemical processes [16] as well as with sampling season [26] and human impacts at downstream sites [25]. In the lake, the high relative abundance of Cyanobacteria dominated by Synechococcus (40.62%) and Microcystis (16.2%) attracted considerable attention due to the high risk of potential toxin producers [38] and the subsequent influence on drinking water supply, aquatic biodiversity, fisheries, water irrigation, recreation, and aesthetics [39].

Co-occurrence patterns of organisms were evaluated to assess community assembly rules and interactions in highly complex systems [40]. Co-occurrence network results revealed that bacterial networks in Erhai watershed had non-random, scale-free, and "small world" properties. The high degree of connectedness and the relatively high proportion of negative associations in the lake bacterial network indicated stronger competition relationships of OTUs in Erhai Lake. Moreover, Proteobacteria play the most important role as a transitional species from streams to lake (Figure 5). Several OTUs of Proteobacteria showed widespread connections with OTUs of lakes and streams bacterial networks. Those broad connections may serve as important connectors that strongly contribute to community stability and coexistence. Removal of those strongly connected components in a network often causes the collapse of structure and function in the ecosystem [40–42]. The topological parameters provided important information that helps us to understand community structure. Higher values of network centralization and heterogeneity indicated stronger connections of OTUs in Erhai Lake (Table 1), implying that a minor disturbance would more easily cause a large impact on the lake bacterial network [41–43]. Moreover, bacterial communities in these networks exhibited strong modular structures, and the stream bacterial networks exhibited a stronger modular structure than other networks (Table 1, modularity values > 0.4). In general, species interactions are more frequent and intense in a modular structure [43]. Notably higher modularity values of the stream networks related to the

heterogeneous habitats were found in streams, which provide abundant niches for microorganisms [40]. Niche differentiation in stream biofilms is also highly associated with heterogeneous microhabitats, wider ranges of physiochemical conditions, and complex hydrological and hydraulic conditions, which are known to have significant effects on composition and function of bacterial communities [44–46]. Thus, heterogeneous microhabitats caused multiple factors affecting bacterial communities in streams.

UPGMA clustering of bacterial communities indicated an apparent spatial variation in bacterial communities in the lake and the streams. In general, physicochemical parameters, nutrients, and nutrient ratios could drive bacterial communities in freshwater ecosystems [34,47–51]. Our results demonstrated that N-factor ($NO_3^-$) (7%) and the interaction of multiple factors, including physicochemical-factor, N-factor, P-factor, and C-factor, drove the bacterial community structure on stream biofilms (Figure 7f). Streams are the primary recipient of xenobiotics and pollutants input from the watershed; thus, terrestrial soil, sewage, plant litter, and autochthonous material were the main sources of organic matter and nutrients in streams [52]. It has been widely reported that agriculture and urban development significantly increase nutrient concentration in streams [53]. These nutrients could accumulate on benthic biofilms and be used by bacterial communities [36,54–56]. In Erhai watershed, streams pass through mountain areas and urbanized areas, which causes significant differences of water quality in these heterogeneity ecosystems under different degrees of human activities perturbations [23]. Under these heterogeneity stream ecosystems, physicochemical-factor provides a diverse environment for bacterial communities [52,57], and DOC, $NO_3^-$, SRP, and TP provide available nutrients for bacterial growth [40,58].

Nutrients predominately explained the variation in lake bacterial communities, possibly due to the relatively uniform physical conditions [4,5]. Elevated inputs of phosphorus and nitrogen from its inflow streams have caused severe eutrophication and associated algae blooms in Erhai Lake. The seasonal succession of Cyanobacteria-dominated communities was highly correlated with nutrient input in the summer at Erhai Lake [51]. Our results demonstrated that P-factor, the interaction of N-factor and P-factor, as well as the interaction of N-factor, P-factor, and C-factor were highlighted as nutrient resources associated with spatial distribution and bacterial species in Erhai Lake during summer (Figures 5 and 7). P-factor (TP and SRP) was positively associated with Cyanobacterial communities (Figure 5), which is understandable because SRP is the most readily available form of phosphorus for autotrophs assimilation [59]. Lake bacterial community structure was positively associated with the SRP concentration in its inflow streams [49]. Thus, phosphorus loading from tributaries from sub-watersheds experiencing seasoning flooding may play an important role in supplying P resources to Erhai Lake. Moreover, low N:P ratios are often connected with an increasing load of P, and the synergy of N and P has historically been the key factor for harmful algal blooms in Erhai Lake [21,60–63]. The most likely mechanism for synergistic effects of N and P is that a single nutrient quickly induces limitation by the alternative nutrient [64]. Additionally, previous studies have shown that organic carbon plays an important role in regulating microbial assemblages and the food web of aquatic ecosystems [65–67]. In the present study, synergistic effects of C-factor and P-factor (C:P) and N-factor and P-factor (N:P) played crucial roles in regulating bacterial communities in Erhai Lake (Figure 7 and Figure S1). Thus, DOC was an important factor affecting bacterial community structures in Erhai Lake that could not be ignored, because high DOC content could provide enough carbon resource for fast-growing and competitive species [64]. TP and SRP were the key environmental factors that drove bacterial communities—especially Cyanobacterial-dominant communities—in Erhai Lake. Moreover, the synergistic effect of C-factor (DOC), N-factor ($NO_3^-$), and P-factor (TP and SRP) also played an important role in regulating the bacterial community structure in the Lake.

## 5. Conclusions

The bacterial communities of Erhai Lake and its tributaries are connected to and affected by terrestrial nutrient enrichment. The lake microorganism network showed stronger centralization and greater heterogeneity than the stream bacterial network, suggesting that lake bacterial communities

were less stabile and resistant to human activities. In contrast, modularity values were higher in streams, implying the abundant niches of microorganisms under different degrees of human activities perturbation. The dramatic differences of bacterial community structure between the lake and the streams were mainly associated with SRP, DOC, C:P, and C:N. Through CCA and VPA analyses, we found that multiple factors, including physicochemical-factor, N-factor, P-factor, and C-factor, were the primary correlates of bacterial communities at the stream scale. However, the composition of lake bacterial communities was most closely associated with phosphorus concentrations and the interaction of N-factor, P-factor, and C-factor (C:P and N:P). These nutrient factors play an important role in structuring bacterial assemblages, stimulating rapid bacterial growth, and producing the Cyanobacteria blooms of Erhai Lake. Our results provide a further understanding of stream–lake linkages from the perspective of bacterial community structures and of the important effects of terrestrial nutrients on downstream ecosystems. These data indicate that it is important to enhance watershed management and decrease the pollution from agriculture and urban regions.

**Supplementary Materials:** The following are available online at http://www.mdpi.com/2073-4441/11/8/1711/s1, Figure S1: Canonical correspondence analysis (CCA) and Variance partition analysis (VPA) of bacterial communities and nutrient ratios in Erhai Lake (a, c, and e) and Streams (b, d, and f). Variance partition analysis (VPA) determined the relative contributions of C:N, C:P, and N:P.

**Author Contributions:** Y.L., X.Q., J.J.E., and M.Z., did the analyses, and prepared the manuscript; J.J.E. reviewed and revised the manuscript. W.P., X.Q., Z.R., and M.Z. designed the study. X.Q., Y.Z., H.Z. and H.Y. performed the field work and laboratory analysis; W.P. and J.J.E. gave suggestions during the whole work.

**Funding:** This study was funded by the State Key Laboratory of Simulation and Regulation of Water Cycle in River Basin (SKL2018CG02), and the National Natural Science Foundation of China (No. 51439007 and No. 41671048), and the IWHR Research and Development Support Program (WE0145B052018 and WE0145B532017).

**Acknowledgments:** We appreciate the anonymous reviewers for their valuable comments and efforts to improve this manuscript.

**Conflicts of Interest:** The authors declare no conflict of interest.

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
