# Peer review of "Impact of Nutrient and Stoichiometry Gradients on Microbial Assemblages in Erhai Lake and Its Input Streams"

_water, doi:10.3390/w11081711_

Round 1

Reviewer 1 Report

L. 52 please Cite in the text and in reference list and in discussion section (L. 411):

Mieczan T., Adamczuk M., Pawlik-Skowrońska B., Toporowska M. 2015. Eutrophication of peatbogs: consequences of P and N enrichment for microbial and metazoan communities in mesocosm experiments. Aquatic Microbial Ecology, 74: 121-141.

Mieczan T., Adamczuk M., Tarkowska-Kukuryk M., Nawrot D., 2016. Effect of water chemistry on zooplanktonic and microbial communities across freshwater ecotones in different macrophyte-dominated shallow lakes. Journal of Limnology, 2:262-274.

L. 86 „microbial community structure” – please precise: bacteria, phycoflora, ect.

L. 193 and 196 please delete “content”

L. 298 please correct and throughout ms “mantel tests” on Mentel tests

Author Response

Point 1:

  L.52 please Cite in the text and in reference list and in discussion section (L. 411):

  Mieczan T., Adamczuk M., Pawlik-Skowrońska B., Toporowska M. 2015. Eutrophication of peatbogs: consequences of P and N enrichment for microbial and metazoan communities in mesocosm experiments. Aquatic Microbial Ecology, 74: 121-141.

  Mieczan T., Adamczuk M., Tarkowska-Kukuryk M., Nawrot D., 2016. Effect of water chemistry on zooplanktonic and microbial communities across freshwater ecotones in different macrophyte-dominated shallow lakes. Journal of Limnology, 2:262-274.

Response 1: These references have been cited in text (line 53, line 400, and line 53).

Point 2: L. 86 microbial community structure” – please precise: bacteria, phycoflora, ect.

Response 2: This error have been revised in the manuscript (line 86).

Point 3: L. 193 and 196 please delete “content”

Response 3: This error have been revised in the manuscript (line 193).

Point 4: L. 298 please correct and throughout ms “mantel tests” on Mantel tests

Response 4: This error have been revised in the manuscript (line 178 and line 300).

Reviewer 2 Report

The manuscript, “Impact of nutrient and stoichiometry gradients on microbial assemblages in Erhai Lake and its input streams” shows the physicochemical water quality data and microbial community assemblage from of a sampling event on Erhai Lake and its contributing streams in August 2016. The manuscript is well written, but requires revisions before I would recommend publication.

General comments:

Could the authors briefly discuss why they chose to calculate C:N:P ratios with a mix of dissolved (C) and total (N & P) nutrients instead of using dissolved fractions of N and P in the methods section? TN seems to be primarily driven by NO3 (spearman plot shows very high correlation coefficient - although both NO3 and NH4 could be summed as DIN) whereas SRP seems to represent dissolved P and is highly correlated with TP – especially in the streams. Would the stoich. results be more convincing if the dissolved fraction of the nutrients were used instead of totals?

The author’s choice of variables included in CCA and VPA do not seem fully appropriate. I appreciate the use of the monte carlo/ mantel tests to reduce final variables included in the models, but some of the final variables selected display significant collinearity (i.e., multicollinearity is found throughout the dataset) – as observed in the spearman plot. For example, the DOC:TP and TP are highly (> |0.8| coefficient) correlated in the lake yet are used in combination in the CCA analysis in Fig. 7a. Can the authors consider removing colinear variables from their CCA and VPA analyses? Alternatively, the authors could use the components of a PCA analysis as variables in their CCA and VPA analyses – this would allow multiple colinear variables to be represented in the CCA and VPA analyses without removing colinear variables within the dataset collected.

Removing collinearity could ultimately cause the authors to choose between nutrient concentrations and nutrient ratios if the PCA solution is not used; however, I would encourage the authors to present similar analyses (by CCA and VPA) with both nutrient concentrations and nutrient ratios – even if just in the supplemental material. I would also encourage the authors to not “mix” concentrations and ratios in the same analyses/plots because of the collinear nature of the variables. This may also help the authors clearly show where (streams or lakes) concentrations and ratios are most predictive of the microbial community composition.

Should DO and altitude be presented in the manuscript? DO data seem to indicate that hypoxia or anoxia was not present in any of the samples collected (Table 1) – Yet DO was included in the CCA analysis in Fig. 7b. Is the DO “signal” in the CCA analysis in Fig. 7b really DO, or is simply less Cond and DOC? I realize that these variables can be related, but the consistency of oxic DO concentrations across all sites seems to indicate that DO is likely not driving the patterns observed.

The collinearity of the statistical analyses seem to cause erroneous conclusions in the discussion section. For example, the paragraph in lines 389 to 406 states that “TN, NO3-, and TP” were dominant factors in streams – however the TN “signal” noted by the statistical analyses is actually because of the strong relation with NO3- (not in addition to it). The same could be said for DO and conductivity, which seem to have a strong negative correlation (although not explicit in the spearman plot). Can the authors revise the discussion after the collinearity has been removed from the analyses?

Minor comments:

Line 87: Aren’t their 4 questions presented here instead of 3?

Figure 1: Can the nutrient ratio data be presented in 3 categories: numerator limiting, co-limitation, and denominator limiting? The other graphics presented in Figure 1 seems fine as is, but reducing the plotting to 3 categories instead of 4 (or using different colors for different size classes on the plotted concentration points) may also help make the figure more clear to the reader.

Figure 5a: Why aren’t all variables included in the analyses presented – why aren’t DO and DOC/TP included here?

Author Response

Point 1: Could the authors briefly discuss why they chose to calculate C:N:P ratios with a mix of dissolved (C) and total (N & P) nutrients instead of using dissolved fractions of N and P in the methods section? TN seems to be primarily driven by NO3 (spearman plot shows very high correlation coefficient - although both NO3 and NH4 could be summed as DIN) whereas SRP seems to represent dissolved P and is highly correlated with TP – especially in the streams. Would the stoich. results be more convincing if the dissolved fraction of the nutrients were used instead of totals?

Response 1:

    Thanks for your valuable suggestion. This advice we have accepted and revised carefully in the manuscript.Relevant analyses, results, discussion, and conclusion about nutrients ratios were revised in carefully the manuscript (lines 197-201).

    NO3- was the major component of TN in streams (57.09%~93.78%) and the lake (86.18%~96.54%), moreover, TN was primarily driven by NO3- because of significantly positively associations between TN and NO3- in Erhai Lake and streams. However, although SRP was the major component of TP in streams (29.17%~74.44%), SRP only was a small component (15.38%~30.95%) in the lake. Thus, in order to ensure the validity of data and the comparability of data between streams and lake, we choose DOC, NO3-, and TP to calculate C:N:P ratios.

Point 2:The author’s choice of variables included in CCA and VPA do not seem fully appropriate. I appreciate the use of the monte carlo/ mantel tests to reduce final variables included in the models, but some of the final variables selected display significant collinearity (i.e., multicollinearity is found throughout the dataset) – as observed in the spearman plot. For example, the DOC:TP and TP are highly (> |0.8| coefficient) correlated in the lake yet are used in combination in the CCA analysis in Fig. 7a. Can the authors consider removing colinear variables from their CCA and VPA analyses? Alternatively, the authors could use the components of a PCA analysis as variables in their CCA and VPA analyses – this would allow multiple colinear variables to be represented in the CCA and VPA analyses without removing colinear variables within the dataset collected.

    Removing collinearity could ultimately cause the authors to choose between nutrient concentrations and nutrient ratios if the PCA solution is not used; however, I would encourage the authors to present similar analyses (by CCA and VPA) with both nutrient concentrations and nutrient ratios – even if just in the supplemental material. I would also encourage the authors to not “mix” concentrations and ratios in the same analyses/plots because of the collinear nature of the variables. This may also help the authors clearly show where (streams or lakes) concentrations and ratios are most predictive of the microbial community composition.

Response 2:

    Thanks for your valuable suggestion. This advice we have accepted and revised carefully in the manuscript (lines 180-181, lines 183-185, lines 294-299).

    CCA were performed to explore the dominant environmental variables that had significant and independent effects on bacterial communities in Erhai watershed. VPA were performed to determine the relative contributions of these key environmental variables including physicochemical-factor, N-factor (NO3-), P-factor (TP and SRP), C-factor (DOC).

    According to the suggestion of reviewer, we present similar analyses with both nutrient concentrations and nutrients ratios by CCA and VPA (Figure 7 and Figure S1), and some collinear variables have been removed from CCA and VPA (Figure 7 and Figure S1). In the revised manuscript, environmental variables were divided into five categories including physicochemical-factor, P-factor, C-factor, N-factor, and nutrient ratios (Table 1). NO3- was the predominant N-factor which was used in CCA and VPA, because NO3- was the major component of TN in Erhai watershed, and it positively associated with TN (p<0.01, Figure 6). Thus, TN and NH4+ have been removed from CCA and VPA. Moreover, SRP not dominated P-factor because SRP was a small component of TP in the lake, thus, TP and SRP constituted P-factor in Erhai watershed. DOC was the C-factor in Erhai watershed (lines 294-299).

    According to the Monte Carlo test, nutrients (C, N, and P) were the crucial environmental factors that were significantly associated with bacterial communities in Erhai Lake. However, except for nutrients, some physicochemical factors including Temp and Cond were also significantly associated with bacterial communities in streams (Figure 7). Thus, there were tree categories (C-factor, N-factor, and P-factor) in Erhai Lake in CCA and VPA, and four categories (physicochemical-factor, C-factor, N-factor, and P-factor) in streams (Figure 7a3 and 7b3). Moreover, the effect of nutrient ratios on bacterial communities have been explored in CCA and VPA (Figure S1).

Point 3: Should DO and altitude be presented in the manuscript? DO data seem to indicate that hypoxia or anoxia was not present in any of the samples collected (Table 1) – Yet DO was included in the CCA analysis in Fig. 7b. Is the DO “signal” in the CCA analysis in Fig. 7b really DO, or is simply less Cond and DOC? I realize that these variables can be related, but the consistency of oxic DO concentrations across all sites seems to indicate that DO is likely not driving the patterns observed.

Response 3:

    Thanks for your valuable suggestion. The heterogeneity of DO in streams (CV=0.079) is higher than that in Erhai Lake (CV=0.12) (Table 1). These streams pass through mountain areas and urbanized areas, which caused significant differences of water quality in these heterogeneity ecosystems under different degree of human activities perturbations. DO were significant lower in Miju stream than that in other streams (t-test, p<0.01), while Cond were significant higher in Miju stream (t-test, p<0.01).  Thus, the spatial heterogeneity of DO and Cond could cause the spatial variations in bacterial communities in streams.  Moreover, we removed DO from CCA and VPA analyses because DO and Cond were the collinear variables (p<0.01, spearman’s rank correlation).

Point 4: The collinearity of the statistical analyses seem to cause erroneous conclusions in the discussion section. For example, the paragraph in lines 389 to 406 states that “TN, NO3-, and TP” were dominant factors in streams – however the TN “signal” noted by the statistical analyses is actually because of the strong relation with NO3- (not in addition to it). The same could be said for DO and conductivity, which seem to have a strong negative correlation (although not explicit in the spearman plot). Can the authors revise the discussion after the collinearity has been removed from the analyses?

Response 4:Thanks for your valuable suggestion. This advice we have accepted and revised carefully in the manuscript. After some collinear variables were removed from the CCA and VPA analyses, the discussion and conclusion have been revised in the manuscript (lines 381-423).

Point 5: Line 87: Aren’t their 4 questions presented here instead of 3?

Response 5: This error have been revised in the manuscript (lines 86).

Point 6: Figure 1: Can the nutrient ratio data be presented in 3 categories: numerator limiting, co-limitation, and denominator limiting? The other graphics presented in Figure 1 seems fine as is, but reducing the plotting to 3 categories instead of 4 (or using different colors for different size classes on the plotted concentration points) may also help make the figure more clear to the reader.

Response 6:

    Thanks for your valuable suggestion. This advice we have accepted and revised carefully in the manuscript, we have reduced plotting to 3 categories in Figure 1, which make the figure more clearly for reader to understand (lines 283-288, and Figure 1).

    In Figure 6, C:P were significantly associated with DOC and TP, N:P significantly associated with NO3- and TP, as well as C:N just significantly associated with NO3-. We can get that both C:P and N:P showed the co-limitation of numerator and denominator, while C:N was limited by denominator in Erhai Lake. In streams, both C:P and N:P was limited by numerator, and C:N was limited by denominator, because DOC significantly associated with C:P, and NO3- significantly associated wih N:P and C:N.

Point 7: Figure 5a: Why aren’t all variables included in the analyses presented – why aren’t DO and DOC/TP included here?

Response 7:

    Thanks for your valuable suggestion. After we redefined C:N (DOC:NO3-), C:P (DOC:TP), and N:P (NO3-:TP), some related results were recalculated and reanalyzed including network analyses (Figure 5 and Table 2). C:P was the key environmental variables which significantly associated with OTUs after network reanalysing.

    Network analyses were conducted to reveal co-occurrence patterns of the microeukaryotic communities. All pairwise Spearman’s rank correlations between those OTUs and environmental variables were calculated. Nodes represent the OTUs and edges represent Spearman’s correlation relationships. Only robust (r>0.6 or r<–0.6) and statistically significant (P-values < 0.01, P-values were adjusted using Benjaminin-Hochber method) were considered in the network (lines 163-167). After spearman’s rank correlation calculation, environmental variables which not significantly associated with OTUs were removed in network visualization analyses. Thus, there were not all variables included in the network analyses (Figure 5). 

    Relevant analyses, results, discussion, and conclusion about network analyses were revised in carefully the manuscript (lines 245-270 and lines 257-378).

Round 2

Reviewer 2 Report

The authors have made changes that have improved and clarified the manuscript. There is only 1 minor comment/edit needed before publication. The review thanks the authors for their revisions and quick turnaround.

Fig. 3c – The authors may consider reducing the number of plotted categories to 3 (as with the other figures), as it is difficult to see the differences in 5 categories on the figure.

Author Response

Point 1: Fig. 3c – The authors may consider reducing the number of plotted categories to 3 (as with the other figures), as it is difficult to see the differences in 5 categories on the figure.

Response 1:

Thanks for your valuable suggestion. This advice we have accepted and revised carefully in the manuscript.We have reduced the number of plotted categories from 5 to 3 in Figure 3c.
